# Circularization restores signal recognition particle RNA functionality in *Thermoproteus*

André Plagens[1], Michael Daume[1], Julia Wiegel[1], Lennart Randau[1,2]*

[1]Max Planck Institute for Terrestrial Microbiology, Marburg, Germany; [2]LOEWE Center for Synthetic Microbiology, Synmikro, Marburg, Germany

**Abstract** Signal recognition particles (SRPs) are universal ribonucleoprotein complexes found in all three domains of life that direct the cellular traffic and secretion of proteins. These complexes consist of SRP proteins and a single, highly structured SRP RNA. Canonical SRP RNA genes have not been identified for some *Thermoproteus* species even though they contain SRP19 and SRP54 proteins. Here, we show that genome rearrangement events in *Thermoproteus tenax* created a permuted SRP RNA gene. The 5'- and 3'-termini of this SRP RNA are located close to a functionally important loop present in all known SRP RNAs. RNA-Seq analyses revealed that these termini are ligated together to generate circular SRP RNA molecules that can bind to SRP19 and SRP54. The circularization site is processed by the tRNA splicing endonuclease. This moonlighting activity of the tRNA splicing machinery permits the permutation of the SRP RNA and creates highly stable and functional circular RNA molecules.

*For correspondence: lennart. randau@mpi-marburg.mpg.de

Competing interests: The authors declare that no competing interests exist.

All organisms contain signal recognition particles (SRPs) that bind to the ribosome and recognize signal peptides in secretory proteins (*Luirink, 2004*). SRP binding can arrest the elongation of nascent peptide chains and targets the ribosome-nascent chain complexes to the endoplasmic reticulum membrane (in Eukaryotes) or the plasma membrane (in Bacteria and Archaea). The SRPs consist of one central SRP RNA molecule and different SRP proteins. Eukaryotes harbor six SRP proteins, Bacteria have one SRP protein and archaeal SRPs are formed by an approximately 300 nt long SRP RNA and the two proteins SRP19 and SRP54 (*Römisch et al., 1989*; *Bernstein et al., 1989*; *Bhuiyan, 2000*; *Cusack et al., 2000*). The SRP RNA is pivotal for the structure and function of all SRPs. Archaeal and eukaryotic SRP RNAs are similar and usually contain eight helical elements that fold into a small Alu-domain and a large S-domain that are connected by an extended linker region (*Batey, 2000*; *Gundelfinger et al., 1983*) (*Figure 1A*). SRP19 binds to the tips of helices 6 and 8 in the S-domain, which form strictly conserved tertiary RNA interactions(*Hainzl et al., 2002*). The SRP RNA adopts a conformation that exposes a SRP54-binding site in helix 8 (h8) (*Rose and Weeks, 2001*) (*Figure 1A*). SRP54 is essential for (i) the recognition and binding of the signal sequence at the ribosome, and (ii) for the GTP-dependent interaction with the SRP receptor of the archaeal plasma membrane (*Luirink et al., 1994*). Intriguingly, few species of the archaeal genus *Thermoproteus* (e.g. *Thermoproteus tenax* and *Thermoproteus uzoniensis*) do not contain canonical SRP RNAs and SRP RNA genes are not annotated (*Siebers et al., 2011*). Genes encoding standard SRP19 (TTX_2083) and SRP54 (TTX_0615) proteins are present and signal sequences can be identified for secretory proteins (*Supplementary file 1A*). Thus, we aimed to search for the elusive SRP RNA genes in *T. tenax* to investigate potential alterations of one of the most conserved ribonucleoprotein complexes found in nature.

We isolated small RNA molecules from *T. tenax* cells and subjected them to Illumina Hiseq2000 RNA-Seq analysis. 13.45 million sequence reads were mapped to the reference genome to obtain

**eLife digest** Cells make many proteins that are eventually released outside the cell or inserted into the cell's membrane. As these proteins are still being made, they are captured by a "signal recognition particle" (or SRP); this molecular machine then guides the newly forming protein to the cell's membrane. SRPs are found in all living organisms on Earth and contain several different proteins and a short RNA molecule. However, a few species belonging to the archaeal domain of life did not seem to contain an identifiable gene for the RNA component of the SRP.

Now Plagens et al. have sought to solve the mystery of the "missing" component of this essential protein-targeting machine. This involved searching through the RNAs that are produced by an archaeon called *Thermoproteus tenax*, a single-celled microbe which grows in the absence of oxygen and at temperatures of up to 95°C.

Plagens et al. discovered that the "missing" SRP RNA gene had not yet been identified because rearrangements in this archaeon's genome had swapped the left and right portions of the SRP RNA gene. Further experiments revealed that the correct sequence order is restored in mature SRP RNA molecules by the two ends of the molecule being linked to form a circle. These RNA circles are made by the cellular machinery that normally removes the unneeded sections from other RNA molecules (called transfer RNAs).

Circular RNA is much more stable at high temperatures and does not degrade easily, and Plagens et al. suggest that this particular arrangement is therefore especially advantageous for this species. Future work will now aim to work out which selective pressures favor the evolution of such fragmented RNAs.

insights into the RNome of *T. tenax*. Subsequently, we searched the sequence reads for minimal conserved SRP RNA features. SRP RNA molecules are highly structured and contain several conserved structural features but only few absolutely conserved sequence segments. These sequences include a conserved SRP54 binding motif in helix 8b and a GNAR tetraloop motif connecting helix 6. We identified these short motifs in an intergenic region between the genes TTX_0683 (pyruvate phosphate dikinase [ppdK]) and TTX_0684 (protein phosphatase [ppa]). The prediction of RNA folding for this intergenic sequence revealed a high potential for extended helical regions, which prompted us to analyze this region in more detail. We utilized the obtained RNA-Seq reads to identify the 5'- and 3'-termini of a 309 nt long highly structured RNA transcript. The alignment of this RNA molecule with other archaeal SRP RNA molecules that are annotated in the SRP database (*Rosenblad, 2003*) highlighted the presence of several conserved structural features (*Figure 1B*). However, the location of the 5'- and 3'-termini of this newly discovered SRP RNA was shifted from the standard location in helix 1 in the small Alu-domain to a region close to a functionally important loop in the large S-domain. This indicates that *Thermoproteus* species with 'missing' SRP RNA genes contain permuted SRP RNA sequences in which the transcripts' termini have been shifted by over 100 nucleotides and the order of sequence motifs has been rearranged. We identified a similar permuted SRP RNA gene in *T. uzoniensis* and noticed that nucleic acid sequence variations do not occur in conserved regions of the SRP RNA and in most cases do not disrupt the structure of the SRP RNA (i.e. G-C base pairs are replaced by G-U or A-U basepairs) (*Figure 1—figure supplement 1*). Sequence reads that mapped to the termini regions of the *T. tenax* SRP RNA revealed unmapped parts consisting of sequences of the distant end, which indicates that these reads span ligated 5'-and 3'-termini (*Figure 2A*). The observed ligation sites are identical and restore the loop at the tip of helix 8 in the permuted SRP RNA. Thus, mature SRP RNAs exist as circular molecules. The RNA-Seq reads also revealed that SRP RNA precursor transcripts contain a 23 nt long leader sequence upstream of the ligated mature 5'-termini (*Figure 2B*). A TATA box promoter element (5'-TTAATA-3') is present 26 nt upstream of the transcription start site in both *T. tenax* and *T. uzoniensis*. Additionally, we detected a C/D box sRNA gene adjacent of the permuted SRP RNA gene (*Figure 2B*). Northern Blot analyses verified the presence of the SRP in *T. tenax* total RNA preparations, and revealed three distinct bands that were reproduced with two different probes (*Figure 2C*). Full-length in vitro transcripts of the SRP RNA (linear and circularized with T4 RNA ligase) were run as size markers. The

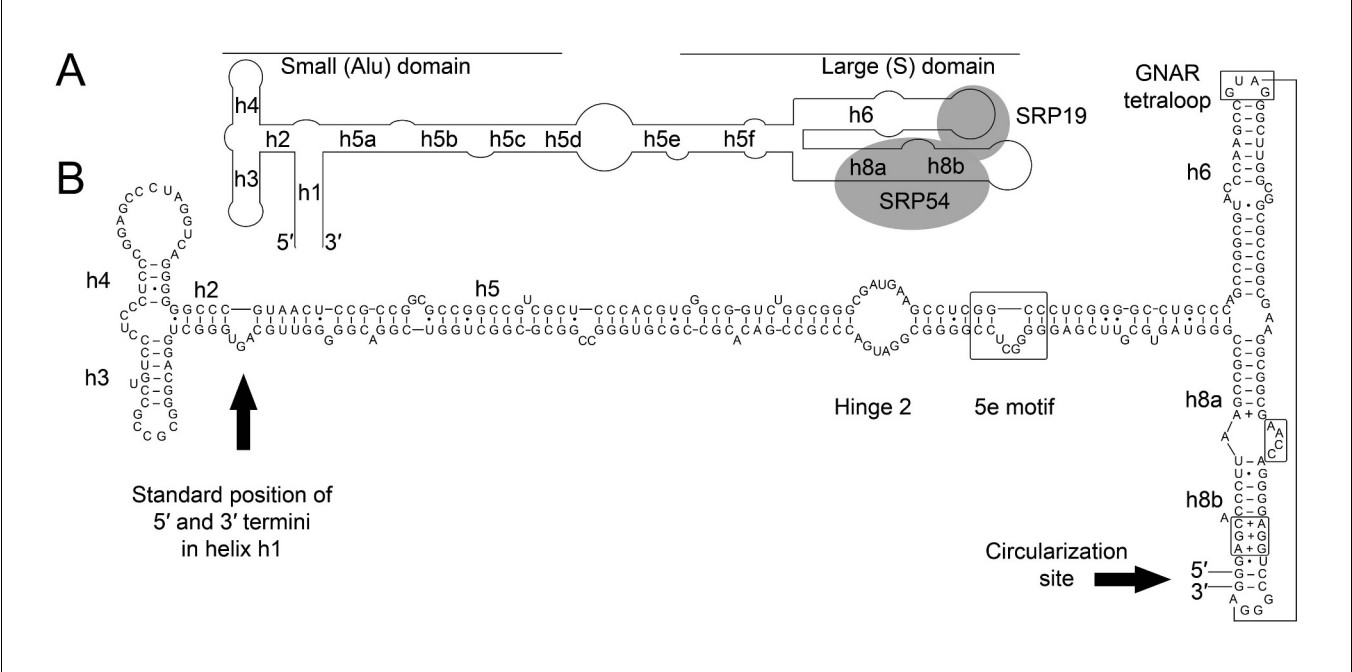

**Figure 1.** Secondary structure and sequence of the *T. tenax* SRP RNA. (**A**) Schematic representation of the canonical archaeal SRP RNA structure. The approximate location of small and large domains, helices (h1 h8) and binding sites of SRP19 and SRP54 proteins are indicated. SRP RNA termini are located in helix h1 in the small domain. (**B**) Sequence and structure of the permuted SRP RNA in *T. tenax*. Conserved features and tertiary interactions are indicated. SRP RNA termini are located at the tip of helix h8 in the large domain.

The following figure supplement is available for figure 1:

**Figure supplement 1.** Conservation of permuted SRP RNAs.

observed migration of this highly structured transcript was identical to the total RNA signal and suggests that the two prominent bands represent the circular form of the SRP RNA (*Figure 2C*).

Next, we aimed to show that the permuted SRP RNA is able to bind SRP19 and SRP54 (*Figure 3A*). Recombinant SRP proteins were produced in *Escherichia coli* and variant S-domains of the SRP RNA were transcribed in vitro (*Figure 3—figure supplement 1* and *Figure 3—figure supplement 2*). DEAE binding assays supported the presence of standard SRP protein recruitment. The interaction of SRP19 with the S-domain is required for the subsequent binding of SRP54 (*Figure 3A*). The open SRP RNA has a significantly reduced affinity for SRP19 and SRP54 (*Figure 3B*). The mutation of conserved SRP RNA motifs (*Zwieb, 1992*; *Zwieb, 1994*) abolished the binding of SRP19 (GNAR loop mutation) or SRP54 (h8b mutation) (*Figure 3B*). These assays verified that the conserved features of the identified SRP RNA molecule are required for SRP19 and SRP54 recognition and revealed that the lack of SRP RNA circularization impairs SRP protein binding (*Figure 3B* and *Figure 3—figure supplement 2*).

Permutation of the SRP RNA sequence removes conserved termini and creates a novel ligation site. We investigated both sites in more detail. The canonical termini of archaeal SRP RNA molecules are located in helix 1 of the small Alu-domain. This helix is absent in the *T. tenax* SRP RNA. It has been suggested that the main function of this helix is to prevent the unfolding of the SRP RNA under the extreme growth conditions of hyperthermophiles (*Zwieb, 2005*). Thus, the replacement of this helix with a continuous sequence in a circular RNA molecule would provide improved RNA stability and a potential evolutionary advantage. The high stability of circular RNA molecules is also a likely explanation for the observed low RNA-Seq coverage as only few SRP RNA fragments were amenable to adapter ligation during RNA-Seq library preparation.

The occurrence of RNA termini near the tip of helix 8 is more problematic for the functionality of the SRP RNA. Ligation of these RNA ends guarantees the presence of a canonical loop in this region.

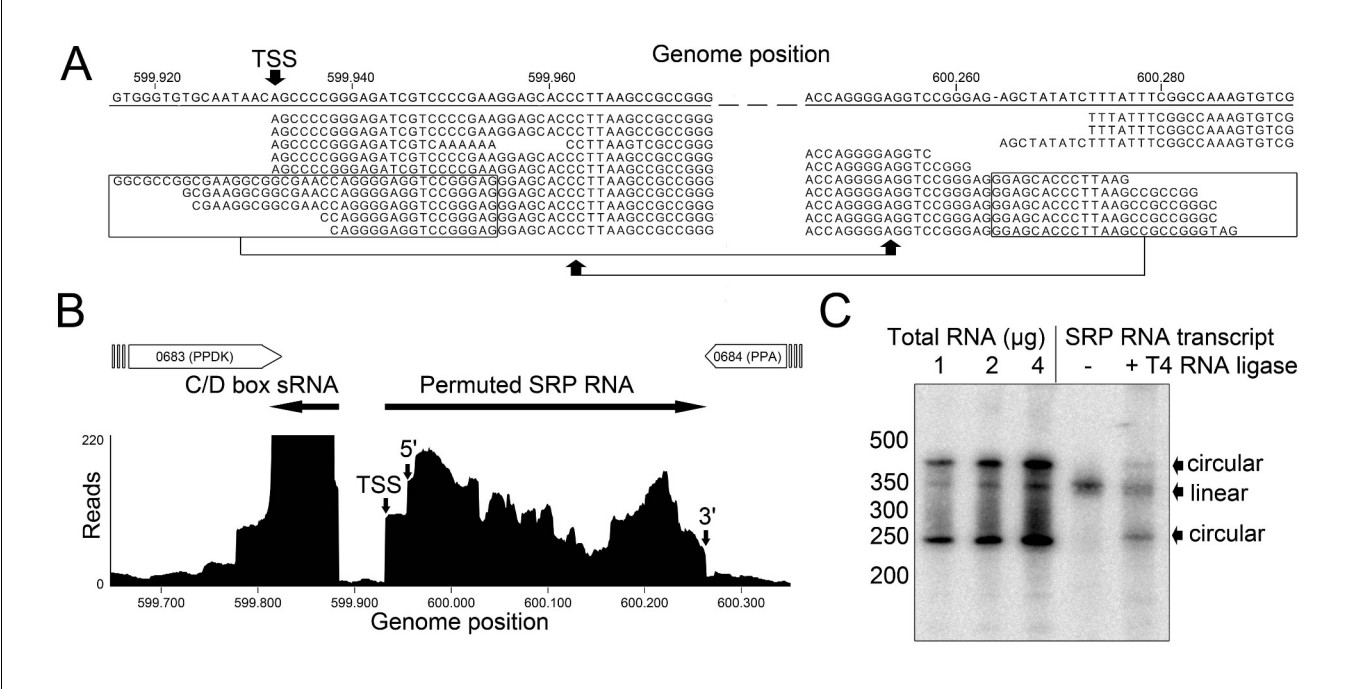

**Figure 2.** Identification of a permuted circular SRP RNA. (**A**) Illumina Hiseq2000 example reads are mapped to the *T. tenax* reference genome (top sequence) and indicate the transcription start site (TSS) of SRP RNA precursors or contain the circularization junction. Parts of these reads (boxed) map to the distant SRP RNA terminus and highlight ligated 5′ and 3′ ends in helix h8b. (**B**) Sequence coverage at the intergenic region between the genes TTX_0683 and TTX_0684. Reads were obtained for the permuted SRP RNA and an adjacent C/D box sRNA (>10000 reads). (**C**) Northern Blot analyses verify the presence of the permuted RNA. Three signals were identified that might represent different stable structures of SRP RNA molecule. 15 ng of linear or circularized SRP RNA transcripts serve as size-markers and verify the altered running behavior of different SRP RNA forms.

However, this ligation reaction requires the presence of clearly defined termini as all sequenced circularization junctions were found to be identical. Thus, we investigated if the observed precursor transcripts might be subjected to endonucleolytic processing to generate mature RNA ends. We observed that the 23 nt long leader sequence of the primary transcript can fold with extended trailer sequences into a so-called bulge helix bulge (BHB) motif (*Figure 4A*). A similar motif is also present at the termini of the permuted *T. uzoniensis* SRP RNA (*Figure 4—figure supplement 1*). BHB elements are structural motifs found in intron-containing transfer RNAs (tRNAs) (*Thompson and Daniels, 1988*). A BHB consists of a central four base pair helix that is flanked by at least one three nt unpaired bulge. These structures are recognized by the tRNA splicing endonuclease that cleaves the bulge sequences and recruits the tRNA ligase RtcB to join the tRNA sequences and circularize the intron sequence (*Popow et al., 2011*; *Englert et al., 2011*). The presence of a BHB motif suggests that RtcB could be responsible for SRP RNA circularization, if the *T. tenax* splicing endonuclease specifically recognized the identified BHB motif. It should be noted, that *T. tenax* contains a record number of 48 introns in 46 tRNAs (*Chan and Lowe, 2009*). These introns are processed by a tRNA splicing endonuclease that belongs to a family of heterotetrameric enzymes (*Randau et al., 2005*). This family shows the most relaxed BHB substrate recognition mechanism and does not rely on the recognition of tRNA sequences for cleavage (*Yoshinari et al., 2009*). We produced recombinant *T. tenax* splicing endonuclease (TTX_1594, catalytic subunit and TTX_1893 structural subunit, *Figure 3—figure supplement 1*) and could verify specific cleavage of the proposed BHB motif (*Figure 4B*). This suggests that the tRNA splicing machinery (i.e. tRNA splicing endonuclease & tRNA ligase RtcB) has a moonlighting activity and processes and circularizes permuted SRP RNA molecules.

Finally, we asked why the SRP RNA gene was permuted in some *Thermoproteus* species and believe that this observation correlates with the emergence of increased tRNA gene fragmentation (*Fujishima et al., 2010*; *Sugahara et al., 2008*; *Randau et al., 2005*). It has been

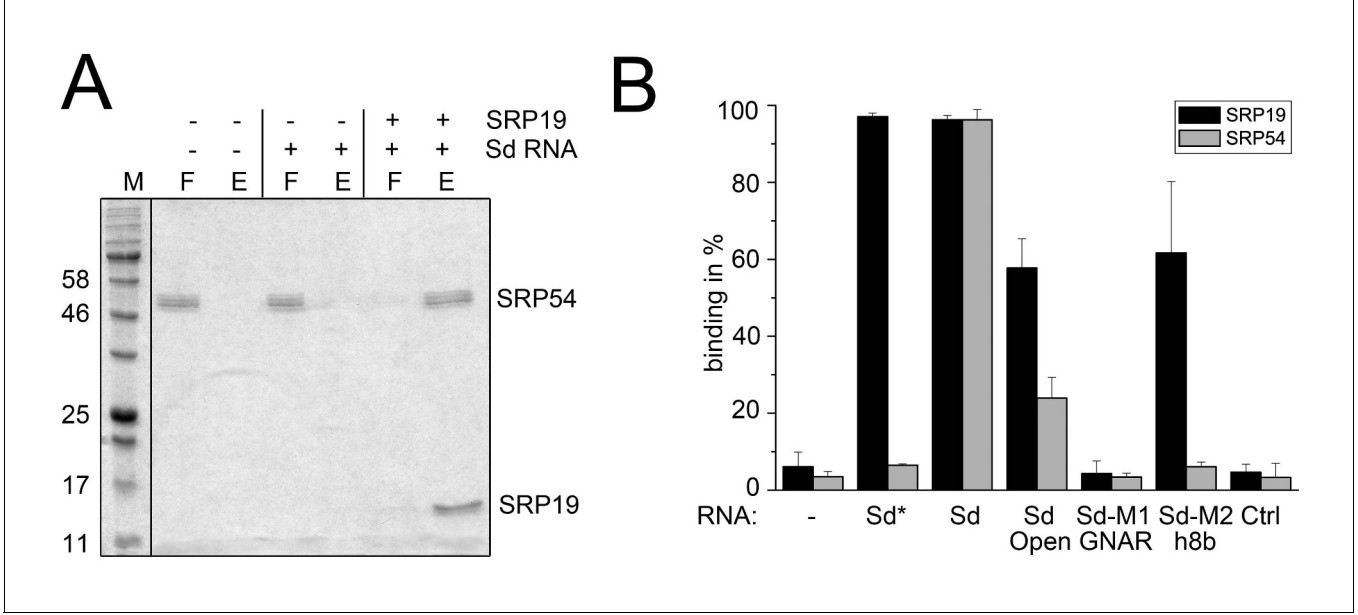

**Figure 3.** Binding of SRP19 and SRP54 to the S-domain of the permuted SRP RNA. (**A**) Proteins of flow-through (F) and elution (E) fractions of DEAE-columns are separated on 15% SDS polyacrylamide gels next to a protein marker (M). SRP54 has weak affinity to the S-domain (Sd) RNA. The addition of SRP19 triggers interaction of SRP54 with Sd RNA. (**B**) Analysis of three independent binding assays of SRP19 and SRP54 to the different S-domain constructs (Sd: S-domain, Open: non-circularized version of the S-domain, GNAR: point-mutation in the GNAR motif, h8b: triple mutation in the helix 8b, Ctrl: control RNA of similar length). Sequences are listed in Tab. S2. Binding of the individual proteins is indicated as Sd*.

The following figure supplements are available for figure 3:

**Figure supplement 1.** Purified recombinant *T. tenax* SRP proteins and splicing endonuclease.

**Figure supplement 2.** Binding of SRP19 and SRP54 to SRP RNA variants.

argued that the fragmentation of tRNAs genes can provide a selective advantage during evolution as tRNA genes frequently serve as viral integration sites (*Randau and Söll, 2008*). The presence of efficient tRNA splicing machinery that can deal with large-scale tRNA intron transposition is required to maintain functionality of a permuted SRP RNA. Thus, it is plausible that permuted SRP RNA genes are derived from an ancestral variant with simple BHB introns. The mechanism of archaeal genome rearrangement events is not known. However, it should be noted that C/D box sRNA genes have previously been observed in the vicinity of split genes and are also located near the permuted SRP RNA gene (*Randau, 2012*). The potential mobility of C/D box sRNA gene sequences will be subject of future investigations. In conclusion, genome rearrangement events near non-coding RNA genes can result in novel maturation pathways of universal RNA molecules. One example is the observed occurrence of permuted and circularized SRP RNA molecules whose increased stability should provide a selective advantage.

## Materials and methods

### Cell culture and RNA preparation

Cells of *T. tenax* Kra1 (DSM 2078) were a kind gift of R. Hensel (Essen) and grown heterotrophically in *Thermoproteus* medium (*Brock et al., 1972*). For the preparation of *T. tenax* total (>200 nt) and small RNAs (<200 nt), 0.1 g pelleted cells were lysed by homogenization and the RNA was subsequently isolated using the mirVanaTM miRNA Isolation Kit (Ambion, Germany) according to the manufacturer's instructions. To generate the SRP S-domain RNA constructs (WT, Open, GNAR, h8b), two forward and reverse complementary DNA oligonucleotides with 15 bases overhangs were synthesized (Eurofins MWG Operon, Germany *Supplementary file 1B*). The oligonucleotides were

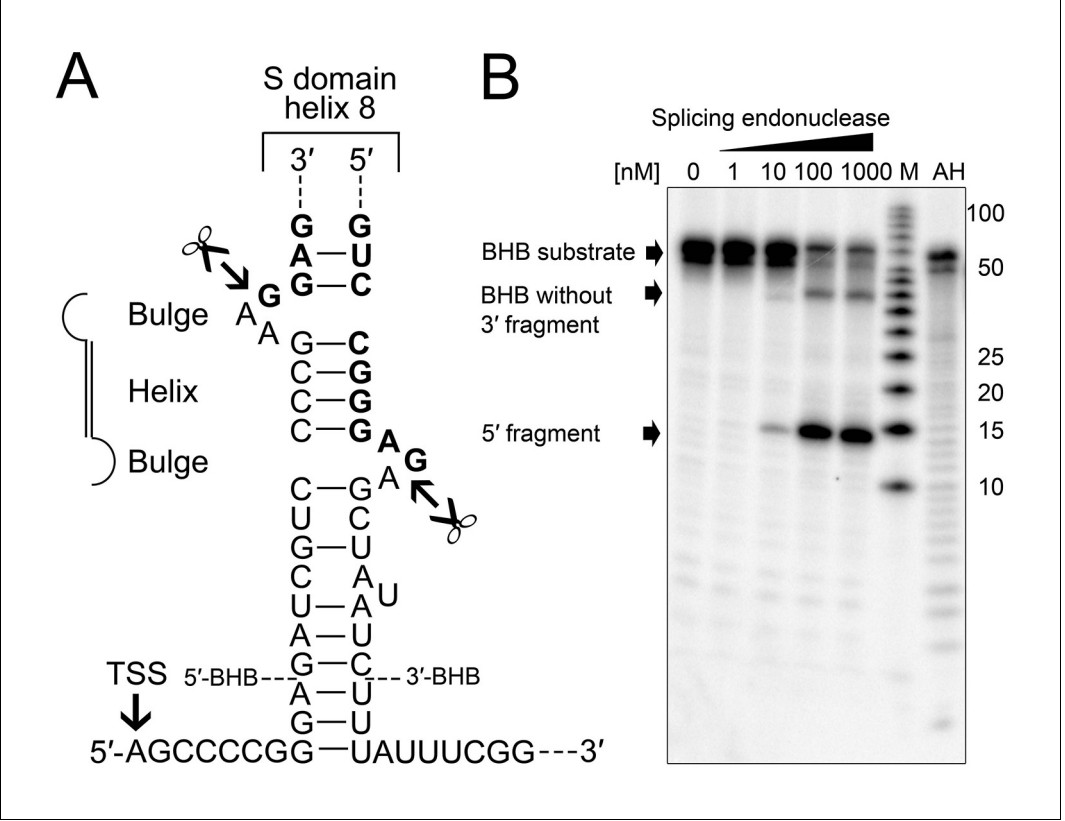

**Figure 4.** Processing of the SRP RNA circularization site by the *T. tenax* tRNA splicing machinery. (**A**) Structure of the proposed BHB motif at the junctions of the leader and trailer sequences (normal letters) and the SRP RNA (bold letters). Cleavage sites are indicated. (**B**) Increasing concentrations of recombinant *T. tenax* splicing endonuclease reveal a distinct processing pattern of 5'-radioactively labeled BHB substrates of the SRP RNA precursor. A label size marker (M) and alkaline hydrolysis (AH) ladder are included. The BHB substrate termini are indicated in *Figure 4A* and the internal sequence is closed by a 3 bp stem and GNAR loop (*Supplementary file 1B*).

The following figure supplement is available for figure 4:

**Figure supplement 1.** A BHB motif is present at the termini of the permuted *T. uzoniensis* SRP RNA gene.

phosphorylated, hybridized and cloned upstream of a T7 RNA polymerase promoter into pUC19. The full-length SRP RNA gene was synthesized with a T7 RNA polymerase promoter and cloned into pMA-RQ (Life Technologies, Germany). The SRP and S-domain RNA constructs were prepared by run-off transcription of 1 µg linearized plasmid as template DNA in a buffer containing 40 mM HEPES/KOH pH 8.0, 22 mM MgCl$_2$, 5 mM dithiothreitol (DTT), 1 mM spermidine, 4 mM of ATP, CTP, GTP and UTP, 20 U RNase inhibitor and 1 µg T7 RNA polymerase at 37°C for 4 hr. The BHB substrate for RNA cleavage assays was synthesized (Eurofins MWG Operon, *Supplementary file 1B*). All RNA molecules were purified by phenol/chloroform extraction (pH 5.2), EtOH precipitated, separated by denaturing-PAGE (8 M Urea, 1× TBE, 10% polyacrylamide) next to a RNA marker (low range ssRNA ladder, NEB, Germany) and visualized by toluidine blue staining. The gel bands were cut out and eluted overnight on ice in 500 µl elution buffer (20 mM Tris/HCl pH 7.5, 250 mM sodium acetate, 1 mM EDTA pH 8.0, 0.25% SDS) and EtOH precipitated.

## Small RNA identification by RNA-Seq

Preparations of *T. tenax* small RNAs were treated with T4 polynucleotide kinase (T4 PNK, Ambion) to ensure proper adapter ligation of RNA termini. The cDNA libraries were prepared according to the TruSeq Small RNA Library Protocol (Illumina) and sequenced using HiSeq2000 technology

(Illumina) at the Max-Planck Genome Centre (Cologne). Sequencing data processing and analyses were performed using CLC Genomics Workbench 8 (Qiagen, Germany). The sequences were trimmed by quality score (limit: 0.05; max. ambiguities: 2), adapter trimming and filtered by length (15 nt cutoff). The trimmed sequences were mapped to the *T. tenax* reference genome (FN869859) using default settings (mismatch cost set to 4). The *T. tenax* RNA-Seq data is available at Gene Expression Omnibus (GSE72127, [http://www.ncbi.nlm.nih.gov/geo/query/acc.cgi?acc = GSE72127]).

## Northern blot

Fractionation of 1–4 µg *T. tenax* total RNA was performed by electrophoretic separation on 8% denaturing-polyacrylamide gels next to an RNA and DNA marker (low molecular weight DNA ladder, NEB). Additionally, to verify the circularization of SRP RNA, 15 ng of full-length SRP RNA in vitro transcript and circularized SRP RNA served as a running control. To obtain a circular SRP RNA control, full length SRP RNA transcript was 5'-dephosphorylated for 6 hr at 37°C using alkaline phosphatase (NEB) and the reaction was stopped by heating for 10 min at 65°C. Then, the RNA was 5'-phosphorylated using T4 PNK (NEB) and ATP for 1 hr at 37°C and ligated using T4 RNA ligase 1 (NEB) for 1 hr at 37°C. The reaction was stopped by incubation at 95°C for 5 min in 2× formamide loading buffer (95% formamide, 5 mM EDTA pH 8.0, 2.5 mg bromophenol blue, 2.5 mg xylene cyanol).

The gel was SYBR Gold stained (Life Technologies) and bands were visualized via UV irradiation. The RNA was blotted by capillary transfer onto nylon membranes (Roti-Nylon plus, Roth) and immobilized by UV crosslinking. Hybridization was performed at 42°C for 18 hr in ULTRAhyb-Oligo buffer (Ambion) with two 5'-labeled probes (the label was added using T4 PNK [NEB] and γ-[$^{32}$P]-ATP [5000 ci/mmol, Hartmann Analytic]) that were complementary to different regions in the SRP S-domain (42 or 45-mer probe, *Supplementary file 1B*). The blot was washed once (2× SSC and 0.1% SDS) at 42°C for 30 min followed by a second washing step (1× SSC and 0.1% SDS) at equal conditions. Radioactive signals were visualized by phosphorimaging.

## SRP RNA binding assays with recombinant SRP19 and SRP54

The *T. tenax* genes *srp19* (TTX_2083) and *srp54* (TTX_0615) were cloned into pET20b creating a fusion with a C-terminal 6× His-tag. The plasmids were transformed into *E. coli* strain DH5α (Invitrogen) and protein production carried out in *E. coli* Rosetta2(DE3)pLysS cells (Stratagene). Cultures were grown in LB medium at 37°C, shaking at 200 rpm. For protein production, 1 mM IPTG was added to a growing culture (OD600: 0.6) which was incubated for 3 hr. SRP19-His expression cells were homogenized in buffer 1 (50 mM Tris/HCl pH 8, 300 mM NaCl), lysed by sonication and cleared by centrifugation (45,000 × g, 1 hr, 4°C). The protein was heat-precipitated (30 min, 70°C) and centrifuged (14,000 × g, 30 min, 4°C). The SRP19-His protein was further purified by Ni-NTA affinity chromatography (HisTrap HP, GE Healthcare) and eluted with a linear imidazole gradient (0–500 mM) at 400 mM imidazole using a FPLC Äkta system (GE Healthcare). SRP54-His was purified by cell lysis in buffer 2 (100 mM potassium phosphate pH 7.5, 500 mM NaCl, 10% glycerol, 1mM ß-Me), lysed by sonication and cleared by centrifugation (45,000 × g, 1 hr, 4°C). The protein was loaded onto a Ni-NTA affinity chromatography column and eluted with a 500 mM imidazole step. Protein fractions were dialysed in buffer 3 (50 mM Tris/HCl pH 7.5, 10% glycerol, 1 mM DTT) and purified over a HiTrap Heparin Sepharose HP column (GE Healthcare) eluting with a linear salt gradient (0–1 M) at 650 mM NaCl.

The formation of SRP RNA-protein complexes was tested using a DEAE-Sepharose assay as described earlier (*Bhuiyan, 2000*; *Gowda, 1997*). 2 µM of the SRP S-domain RNA constructs (WT, Open, GNAR, h8b) were folded in 55 µl binding buffer (50 mM Tris/HCl pH 8, 300 mM KOAc, 5 mM MgCl2, 1 mM DTT), heated for 10 min at 65°C and cooled down to room temperature for 1 hr. 2 µM of SRP19 or SRP54 were added and the reaction volume was adjusted to 70 µl. The mixture was incubated for 45 min at 37°C and immediately loaded onto an 80 µl bed-volume DEAE-Sepharose column (GE Healthcare) equilibrated in binding buffer. The flow-through (F) was combined with a 70 µl binding buffer wash. Bound complex (E) was eluted twice with 70 µl of elution buffer (binding buffer with 1 M KOAc). Fractions F and E were TCA-precipitated, separated on 15% SDS-

polyacrylamide gels, stained with InstantBlue and analyzed with the ImageJ software (imagej.nih. gov/ij/). The binding efficiency was tested in three independent experiments.

## RNA cleavage assays with recombinant tRNA splicing endonuclease

The *T. tenax* genes TTX_1594 (catalytic subunit) and TTX_1893 (structural subunit) were cloned together into pETDuet-1 and the protein was purified as described earlier(*Zwieb, 2005*). Briefly, cells were resuspended in buffer 4 (50 mM Tris/HCl pH 7.5, 500 mM NaCl, 3 mM DTT), sonicated and cleared by centrifugation (45,000 × g, 1 hr, 4°C). The cell lysate was heat-precipitated (30 min, 80°C) and centrifuged (14,000 × g, 30 min, 4°C).

10 pmol of the BHB substrate for RNA cleavage assays was 5'-labeled with T4 PNK (Ambion) and γ-[$^{32}$P]-ATP (5000 ci/mmol, Hartmann Analytic) and gel-purified as described above. 1–1000 nM purified splicing endonuclease was incubated with 2 nM of 5'-labeled RNA in nuclease buffer (40 mM Tris/HCl pH 8, 2 mM MgCl2, 1 mM EDTA) at 60°C for 20 min. The cleavage reaction was stopped by adding formamide loading buffer and incubating at 95°C for 5 min. The reaction was loaded onto a 20% denaturing polyacrylamide gel running in 1× TBE, 6 W for 4 hr next to a low molecular weight marker (10–100 nt, Affymetrix) and an alkaline hydrolysis ladder of the BHB substrate. The cleavage products were visualized by phosphorimaging.

## Acknowledgements

The authors were supported by Max Planck Research Group funding of the Max Planck Society. We thank Gert Bange for helpful advice.

## Additional information

### Funding

| Funder | Author |
| --- | --- |
| Max-Planck-Gesellschaft | Lennart Randau |

The funders had no role in study design, data collection and interpretation, or the decision to submit the work for publication.

### Author contributions

AP, LR, Conception and design, Acquisition of data, Analysis and interpretation of data, Drafting or revising the article; MD, JW, Acquisition of data, Analysis and interpretation of data

## Additional files

### Supplementary files

• Supplementary file 1. Signal peptide analysis and oligonucleotide sequences. This text-file contains two extended tables. Supplementary file 1A contains a list of predicted signal peptides for *T. tenax*. Supplementary file 1B lists all oligonucleotides and RNA sequences that were used in this study for cloning and RNA substrate generation.

### Major datasets

The following datasets were generated:

| Author(s) | Year | Dataset title | Dataset URL | Database, license, and accessibility information |
| --- | --- | --- | --- | --- |
| Daume M, Plagens A, Randau L | 2015 | Identification of the Thermoproteus tenax small Rnome | http://www.ncbi.nlm.nih. gov/geo/query/acc.cgi? acc=GSE72127 | Publicly Available at the NCBI Gene Expression Omnibus (Accession no: GSE72127) |

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
