## [Decision Letter]

Thank you for submitting your work entitled "Circularization restores SRP RNA functionality" for consideration by *eLife*. Your article has been favorably evaluated by James Manley (Senior Editor) and two reviewers, one of whom is a member of our Board of Reviewing Editors.

The reviewers have discussed the reviews with one another and the Reviewing editor has drafted this decision to help you prepare a revised submission.

In this short report, Plagens et al. show that the SRP RNAs of *Thermoproteus tenax* and *Thermoproteus uzoniensis* are encoded in a permuted fashion and that the RNA ends are ligated to form a circular molecule, possibly to confer added stability to the RNA. The authors also show that a tRNA splicing endonuclease generates the site of circularization, pointing to a moonlighting activity for this enzyme. The reviewers only have limited comments:

1) The title is vague and should mention the archaeal focus of the study.

2) Figure 4: The ability of a recombinant protein to act on an RNA does not by itself provide ironclad support for the authors' suggestion that the tRNA splicing machinery plays a role in processing these circular SRP RNA in cells. Can mutational studies be easily performed in living cells to provide stronger support for biological relevance of the authors' conclusion of moonlighting activities of the tRNA processing machinery?

3) While the sequencing data and co-migration with ligated RNAs in Figure 2 is consistent with the conclusion of circular RNA, the authors may wish to add some additional experimental tests (e.g. RNase R resistance) to provide complementary data to firmly establish this key point of the study.

---

## [Author Response]

*In this short report, Plagens et al. show that the SRP RNAs of* Thermoproteus tenax *and* Thermoproteus uzoniensis *are encoded in a permuted fashion and that the RNA ends are ligated to form a circular molecule, possibly to confer added stability to the RNA. The authors also show that a tRNA splicing endonuclease generates the site of circularization, pointing to a moonlighting activity for this enzyme. The reviewers only have limited comments:1) The title is vague and should mention the archaeal focus of the study.*

We changed the title. SRP is spelt out in full and “…in *Thermoproteus*” was added.

*2) Figure 4: The ability of a recombinant protein to act on an RNA does not by itself provide ironclad support for the authors' suggestion that the tRNA splicing machinery plays a role in processing these circular SRP RNA in cells. Can mutational studies be easily performed in living cells to provide stronger support for biological relevance of the authors' conclusion of moonlighting activities of the tRNA processing machinery?*

Unfortunately, genetic systems are not available for *Thermoproteus* species (see e.g. Atomi, H. et al., Overview of the genetic tools in the Archaea, Front Microbiol 2012, 3, 337). We provide additional support with a new figure supplement (Figure 4—figure supplement 1) highlighting the conservation of a bulge-helix–bulge structure in *Thermoproteus uzoniensis.*

*3) While the sequencing data and co-migration with ligated RNAs in Figure 2 is consistent with the conclusion of circular RNA, the authors may wish to add some additional experimental tests (e.g. RNase R resistance) to provide complementary data to firmly establish this key point of the study.*

We performed Northern blot assays with RNase R treated total RNA (1-20U RNase R/μg RNA) and saw no significant reduction of the signal. However, RNase R resistance might also be observed for the highly structured linear SRP RNA with its four nucleotide 3’ overhang as it was shown that “duplex RNAs with no overhang or with a 4-nucleotide overhang bind extremely poorly to RNase R and are inactive as substrates”. (Vincent, H.A. and Deutscher, M.P., Substrate recognition and catalysis by the exoribonuclease RNase R, J Biol Chem 2006, 281, 29769-29775). We sequenced the termini ligation site in hundreds of unique reads in independent RNA-Seq datasets (available at GEO database GSE72127) and consider this to be conclusive evidence.